# Differential diagnosis of a calcified cyst found in an 18th century female burial site at St. Nicholas Church cemetery (Libkovice, Czechia)

**Barbara Kwiatkowska**[1], **Agata Bisiecka**[1]\*, **Łukasz Pawelec**[1], **Agnieszka Witek**[1], **Joanna Witan**[2,3], **Dariusz Nowakowski**[1], **Paweł Konczewski**[1], **Radosław Biel**[1], **Katarzyna Król**[1], **Katarzyna Martewicz**[1], **Petr Lissek**[2], **Pavel Vařeka**[3], **Anna Lipowicz**[1]

1 Division of Anthropology, Institute of Environmental Biology, Wrocław University of Environmental and Life Sciences, Wrocław, Poland, 2 Institute for Preservation of Archaeological Heritage of Northwest Bohemia, Most, Czechia, 3 Department of Archeology, University of West Bohemia in Pilsen, Pilsen, Czechia

\* agata.bisiecka@upwr.edu.pl

## Abstract

During archaeological excavations in burial sites, sometimes stoned organic objects are found, in addition to human remains. Those objects might be of a different origin, depending on various factors influencing members of a community (i.e. diseases, trauma), which provides information about their living conditions. The St. Nicholas Church archaeological site (Libkovice, Czechia) in the 18th century horizon of the cemetery, yielded a *maturus-senilis* female skeleton with a stone object in the left iliac fossa. This object was an oviform cyst-like rough structure, measuring 54 mm in length, 35 mm in maximum diameter and 0.2–0.7 mm shell thickness. Within the object there were small fetal bones (long bones, i.e. femur and two tibias, two scapulas, three ribs, vertebrae and other tiny bone fragments). Methods utilized to analyze the outer and inner surface morphology of the cyst and its inside, included: X-ray, CT imaging, SEM, histological staining and EDS. The EDS analysis revealed the presence of primarily oxygen, calcium and phosphorus in bone samples, and oxygen and silicon, in stone shell. Based on the length of the femur (20.2 mm) and tibia (16 mm) shafts, the fetal age was determined as being in the 15–18 week of pregnancy. The differential diagnosis was conducted, including for the three most probable cases: *fetiform teratoma* (FT), *fetus-in-fetu* (FIF) and *lithopedion*. The possibility of *fetiform teratoma* was discounted due to the presence of an anatomically correct spine, long bones and the proportions of the find. Although the low calcium content in the shell (2.3% atom mass), the lack of skull bones and the better developed lower limbs indicate *fetus-in-fetu* rather than *lithopedion*, the analyses results are unable to conclusively identify the object under one of these two categories since there are insufficient such cases in excavation material with which to draw comparison.

**Data Availability Statement:** All relevant data are within the paper.

**Funding:** The author(s) received no specific funding for this work.

**Competing interests:** The authors have declared that no competing interests exist.

## Introduction

Research on human burials (both in terms of whole populations, as well as single individuals) is a valuable source of information about living conditions, socio-economic status and occurrence of diseases across continents. In addition to biological materials (e.g. human or animal bones), human burials also often contain archaeological artefacts [1]. However, other objects can sometimes be found among skeletal remains, such as calcified or ossified masses and calculi [2, 3]. Evidence of the latter indicates that they occur in human burials worldwide [4]. Such pathologies can appear in different areas of the body for a variety of reasons [5]. Documentation in the medical literature about such finds in living individuals is also systematically increasing. The latter is due to the development and widespread use of imaging methods that allow for the detection of symptomless diseases and conditions that do not pose a direct threat to health or life. However, by contrast, few cases of such calcifications are described in the palaeopathological literature. Not all disease states can be identified in an archaeological context–palaeopathologists have almost only bone material at their disposal, and some diseases may leave no trace on the bone tissue or the traces observed on bones may be non-specific and not lead to a clear diagnosis. Also it may be difficult (or even impossible) to distinguish between peri-mortem and post-mortem changes in archaeological material [6].

In this regard, the calcifications predominantly described in the archaeological literature include: urinary stones [4, 5, 7, 8], tumor calcifications [9–11], calcified parasites [12–14]. Previously discovered stones were predominantly collected from mummies [7, 15, 16]. Descriptions of nodules of the genital organs have also been reported in the palaeopathological literature. The conditions and diseases described can be of different origin and are clinically classified according to their morphological and radiological appearance: benign tumors such as ovary (i.e. gonadoblastoma, teratoma), uterine tumors (i.e. leiomyoma); cancer of the ovary, uterus; vascular calcifications (i.e. arteriovenous malformations, uterine arterial calcifications). If the nodule is localized in the pelvic region, a possible etiology is the ectopic pregnancy (*graviditas extrauterina*), especially when parts of the skeleton are discovered inside the remains [17–20].

A specific type of ectopic pregnancy is a *lithopedion* (gr. *lithos*–stone and *paidion*–child). This condition occurs in 1.5–2.0% of ectopic pregnancies [21–25] which represents approximately 0.0054% of all gestations [26]. The *lithopedion* is an extra-uterine fetus that has become bony or calcified and has remained inside its mother [21, 27–29]. Three types of this rare phenomenon have been observed (mostly in clinical cases), depending on the scope of its calcification [18, 22, 30–32], where calcification involves: fetal membranes, both fetus and membranes, or only the fetus, respectively [23, 33].

*Fetiform teratoma* is a rare type of ovarian tumor resembling a deformed human fetus [34]. It is composed of at least one of the embryonic germ layers and can develop into highly differentiated tissues, creating an organized fetal structure. Teratomas have been documented in archaeological evidence. For example, ovarian teratoma was discovered at La Fogonussa, an archaeological site from the late Roman period [35]. The ovarian teratoma was found in the pelvic region of a 30–40-year-old woman. Another ovarian teratoma was also found in the abdominal cavity of a 15–20-year-old woman from Eten, Peru [36].

*Fetus-in-fetu* (FIF) is a rare condition, often described as an asymptomatic abdominal mass, usually associated with infancy or childhood [37], although there is evidence of FIF having been found in a 47 year old man [38]. With regards to morphology, FIF should exhibit signs of axiation and metameric segmentation, with organs arranged appropriately along the skeletal axis [39, 40].

A human skeleton with a cystic object in its pelvic region was discovered during archeological excavations in Libkovice Village (North-West Czechia). This paper presents the results of

analyses using anthropological, radiological and histological methods. These results could facilitate a differential diagnosis and an identification of the cystic object as either *lithopedion*, FIF or FT, particularly within an archaeological context where such cases are a rarity.

## Materials and methods

### Site and burial description

No permits were required for the described study, which complied with all relevant regulations. St. Nicholas Church cemetery in Libkovice (N 50˚34.75580', E 13˚41.00348') is situated in the upper part of the wide Lom stream valley, 3 km South-East of the foothills of the Ore Mountains (*Krušné hory/Erzgebirge*). Archaeological research suggests that the oldest church in this village was built at the turn of the 12th and 13th centuries [41].

A cemetery was identified at the church where two horizons of burials were observed: first (12–13th century), following which this was perhaps divided into two separated sites (up to the 19th century) [42]. The most recent excavations of the cemetery in Libkovice were carried out in the summers of 2019 and 2020, where two out of the four sectors of the churchyard were explored. In the explored sectors, 199 single graves and 7 ossuaries with the minimal number of individuals (MNI) at 225, were excavated. All the research material has been transferred to the laboratory of the Department of Archaeology, University of West Bohemia in Pilsen (Sedláčkova 38, 306 14 Plzeň 3) for analysis, after which it has been transported to the repository of the Institute for Preservation of Archaeological Heritage of Northwest Bohemia in Most (Jana Žižky 835/9, 434 01 Most). Among those single burials, the remains in grave No. 1095 (sector II, square 15, layer 1094) are the subject of this paper (Fig 1). No specific permissions were required for this research.

The skeleton from this grave was buried in a trapezoidal wooden coffin. A small metal cross, made from copper alloy by casting and fashioned with the figure of a crucified Christ, was discovered between the bones of the left hand. The cross has trifolius shoulder tips and a round loop at the top positioned transversely to the surface of the artefact. This type of cross is the 18th century so-called Latin Cross and is widely known in Czechia and neighboring countries [43, 44]. This discovery established that the burial dates back to the 18th century [42].

### Individual description

The skeleton was well-preserved, arranged on the north-south axis, laid on its back in a horizontal position, in the correct anatomical arrangement. The head was positioned at the north-end with the face directed towards the east. The upper limbs were flexed at the elbow joints, the hands laid at the pelvis, lower limbs straight. The skull was complete (*cranium*), displaying total intravital loss of teeth and lowering of the alveolar process almost to the level of the palate in the case of the jaw (5–7 mm) and to the midpoint of the mandible shaft (15 mm). The spine was also complete, as was the pelvis and limbs. The ribs and sternum were damaged.

In the skeleton's left iliac fossa an oviform structure was discovered and secured for further study. All the osteological material is in the repository of the Institute for Preservation of Archaeological Heritage of Northwest Bohemia in Most. For further research, the found object was transported to the Department of Anthropology, Wrocław University of Environmental and Life Sciences.

### Cystic object macroscopic description

The discovered mass was oviform, stone and rough (Fig 2). The object measured 54 mm in length, 35 mm in maximum diameter and 0.2–0.7 mm in shell thickness. The object was

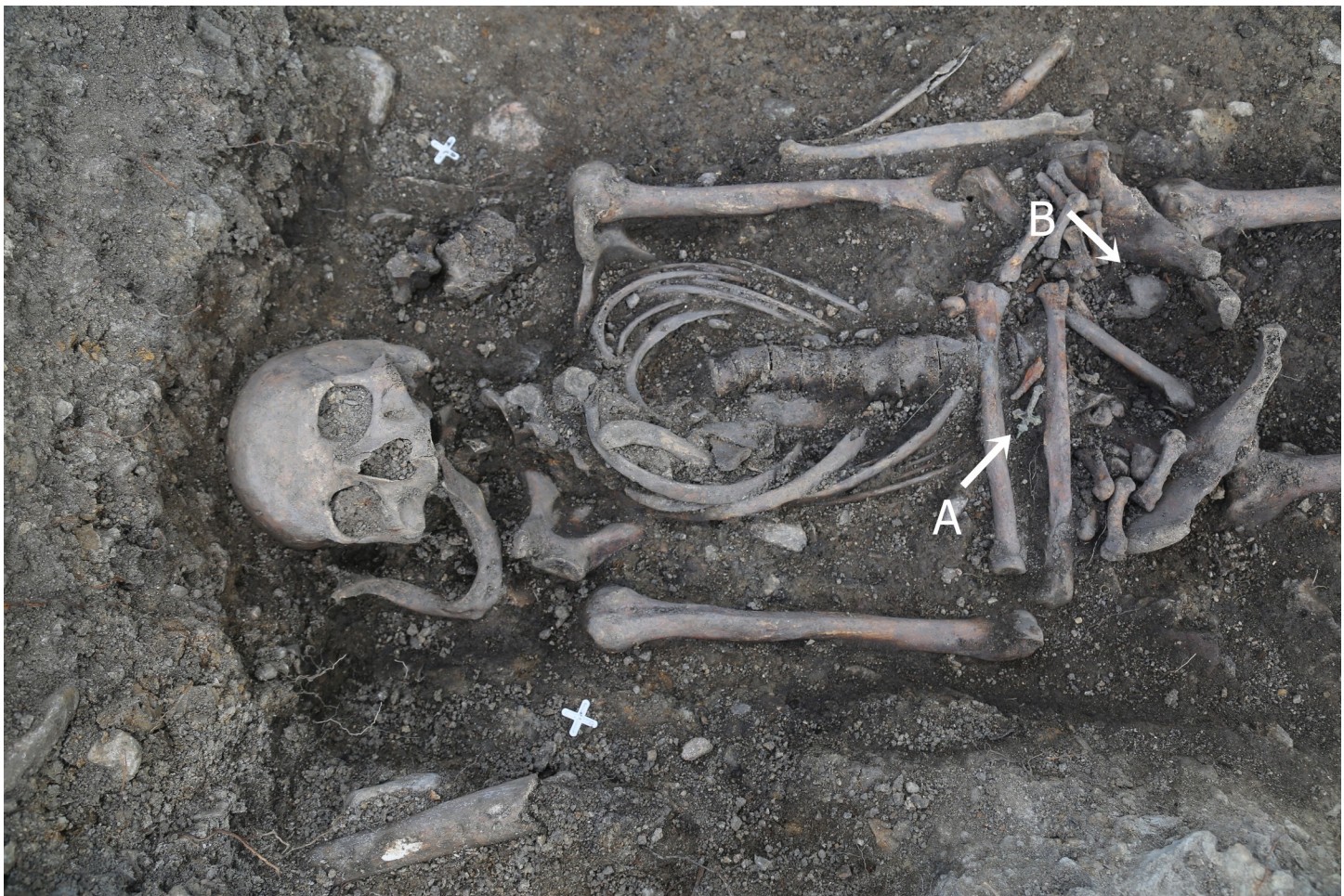

**Fig 1. Burial no. 1095.** A: Latin Cross. B: cystic object in left iliac fossa.

completely covered in soil and partially damaged, probably resulting from depositional processes. The cyst was a compact structure filled with sediment, a mixture of earth and bone-shaped structures.

## Methods

Standard anthropological methods were used to determine the age and sex of the individual [45], the incidence of skeletal pathologies [46] and body height (the Trotter and Gleser method and the Nainys method [47]).

The interior of the cyst was observed using a Nikon SMZ800 stereoscopic microscope with an SHR Plan Apo 1x lens, reflecting white light. Samples of material from the cyst's interior were then collected for histological and chemical analyses. Following dehydration and cleaning in acetone, the histological samples were embedded in methyl methacrylate. Following polymerization, the blocks were trimmed to orient the section plane and provide a complete frontal view of the specimen. A rotating diamond saw was used to extract a cut from each bone (Isomet Low Speed Saw, Buehler, Ltd.) of 2 mm thick slices, which were polished using the diamond papers with a gradation of 600, 1200 and 4000 (MetaServ 250 polishers, Buehler, Ltd.). The slides were analyzed using a Nikon Eclipse 80i microscope, transmitted white light and

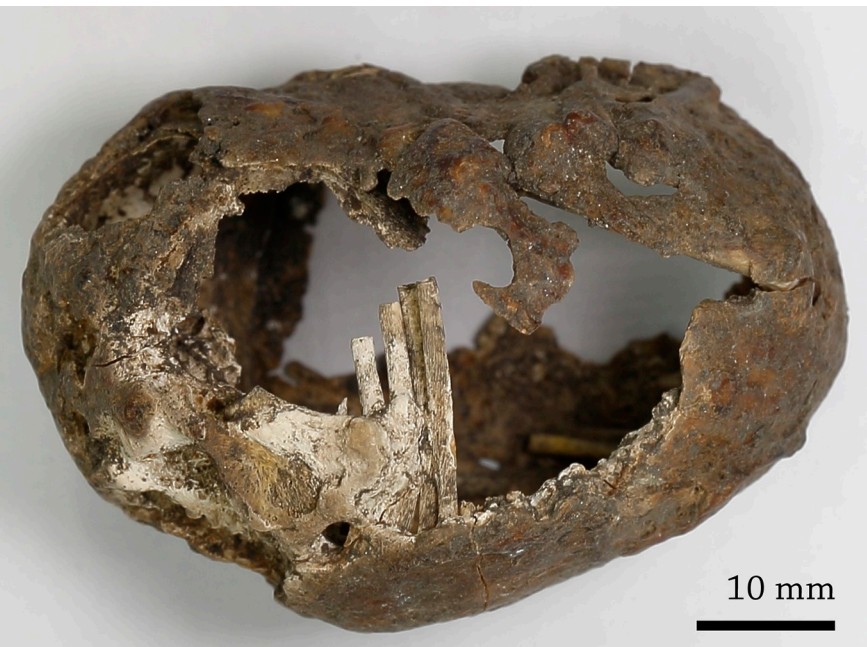

**Fig 2. Outer view of the cyst.**

UV reflected light using Nikon-B2A filter. A Panasonic sensor digital microscope camera U3ISPM with the RisingCam® software was used for photographic documentation and taking measurements.

The SEM (scanning electron microscope) observations were conducted for two samples (bone and shell), which were cleaned in 70% and 100% acetone (each for 24 hours) and in the ultrasonic scrubber. Sputter coating was not utilized during the SEM analysis in order to avoid destruction of the structures. These analyses served to identify irregularities in morphology, as well as to identify the elemental composition of bones and shell.

Energy Dispersive X-ray Spectroscopy (EDS) measurements were performed for selected elements in order to estimate the chemical components of material (both from bone and shell) (Table 1). The analyses were conducted using SEM, EVO LS15, Zeiss with energy dispersive X-ray analysis (SEM/EDX) applying quantax detector (Brüker) with 10 kV of filament tension.

Conventional radiographs were taken with 120 mA and 35 kV. Our own modification was applied, where the distance between the object and the matrix was set at 50 cm, and the distance between the object and the focus was also set at 50 cm. X-Ray analyses were performed in order to determine the skeletal arrangement in the structure, as well as to conduct measurements of selected bones.

The X-Ray imaging was performed using Konica Minolta Regius Model 110s with Siemens Polydoros LX 30 generator.

CT imaging was performed using a Siemens Somatom Emotion 16 (3rd generation) apparatus in 0.6 mm increments.

## Results

### Anthropological analyses of the skeleton

Based on the shape of the surface of pubic symphysis and additionally the obliteration of the cranial sutures, as well as degenerative changes observed in the bone material, associated with

**Table 1. EDS analysis results of the bone and shell.**

| Element | Bone | | | | Shell | | | |
|---|---|---|---|---|---|---|---|---|
| | Mass [%] | Mass Norm [%] | Atom [%] | abs. error [%] | Mass [%] | Mass Norm [%] | Atom [%] | abs. error [%] |
| Aluminium | 0.28 | 0.24 | 0.20 | 0.2 | 10.50 | 10.36 | 7.75 | 3.9 |
| Calcium | 38.30 | 31.96 | 17.81 | 1.2 | 4.62 | 4.56 | 2.30 | 0.2 |
| Carbon | 2.75 | 2.30 | 4.27 | 0.9 | 0.91 | 0.90 | 1.51 | 1.0 |
| Copper | 0.71 | 0.59 | 0.21 | 0.1 | 0.25 | 0.24 | 0.08 | 0.0 |
| Fluorine | 3.88 | 3.24 | 3.81 | 1.7 | 1.35 | 1.34 | 1.42 | 1.9 |
| Iron | 0.52 | 0.44 | 0.17 | 0.1 | 3.38 | 3.33 | 1.20 | 0.1 |
| Magnesium | 0.70 | 0.58 | 0.53 | 0.1 | 1.29 | 1.27 | 1.05 | 0.2 |
| Oxygen | 53.12 | 44.32 | 61.87 | 6.6 | 55.67 | 54.89 | 69.29 | 6.8 |
| Phosphorus | 14.86 | 12.40 | 8.94 | 0.6 | 3.32 | 3.28 | 2.14 | 0.2 |
| Potassium | 0.24 | 0.20 | 0.11 | 0.0 | 1.62 | 1.59 | 0.58 | 0.1 |
| Selenium | 1.79 | 1.50 | 0.42 | 0.2 | 0.61 | 0.60 | 0.15 | 0.1 |
| Silicon | 0.12 | 0.10 | 0.08 | 0.0 | 15.70 | 15.48 | 11.13 | 0.7 |
| Sodium | 1.72 | 1.43 | 1.39 | 0.2 | 1.02 | 1.00 | 0.88 | 0.1 |
| Strontium | 0.85 | 0.71 | 0.18 | 0.1 | 1.18 | 1.17 | 0.27 | 0.2 |
| Total: | 119.84 | 100.00 | 100.00 | – | 101.42 | 100.00 | 100.00 | – |

the aging process (presence of articular cartilage degenerative lesions and degeneration of the vertebrae surface), considering that this woman was from a rural population, we estimated the age of individual at 45–55 years old. The individual was 158 cm tall.

Numerous pathological changes to bones were discovered, mainly consisting of spinal degenerative changes in the form of osteophytes (up to 5–7 mm), and degeneration of the articular processes, especially in the thoracic (Fig 3) and lumbar regions. Moreover, two last vertebrae arches fusion (L4–L5) with lesions of hypertrophic nature and osteochondrosis of L5 and S1 vertebrae bodies, were noted in the lumbar spine. Degeneration of the interphalangeal joints in the hands was also found, probably resulting from advanced age and/or rheumatic diseases. Additionally, significantly developed muscle attachments, especially on femurs, sharp elongated interosseous edges on forearms and lower limbs, may indicate a female's possible physical workload. The presence of the cyst was not related to the aforementioned conditions.

## Cyst analyses

The X-ray examination (Fig 4) showed heterogeneous lesions with discontinuous oval calcification, containing a few more calcified structures. A small number of calcified structures resembling long bones (i.e. femur), ribs and spinal elements were visible. The structure identified as the femur, measured 20.2 mm at maximum length.

CT imaging (Fig 5A–5C) revealed irregular thickness of the object in the cross-section, whilst the mass of the interior revealed varying degrees of X-ray shadow saturation and voids. The outer surface of the cyst was not rough (although there were slight irregularities). The inner surface showed greater irregularity and fine porosity.

In the optical microscopic observation, the outer surface of the cyst was irregular, but not rough (as with the RTG observation). Fetal bones were discovered inside the cyst: femur, tibia (two), scapula (two), ribs (three), vertebrae and multiple tiny, undetermined bone fragments (Fig 5D). Relevant for a differential diagnosis was an absence of bone fragments resembling the skull shape. The maximum length of the tibias was photometrically measured (16 mm). The surface of the scapula and ribs were very fragile, with poorly developed bone compact tissue (Fig 6). The SEM analysis showed the same inner and outer surface shape, as revealed in

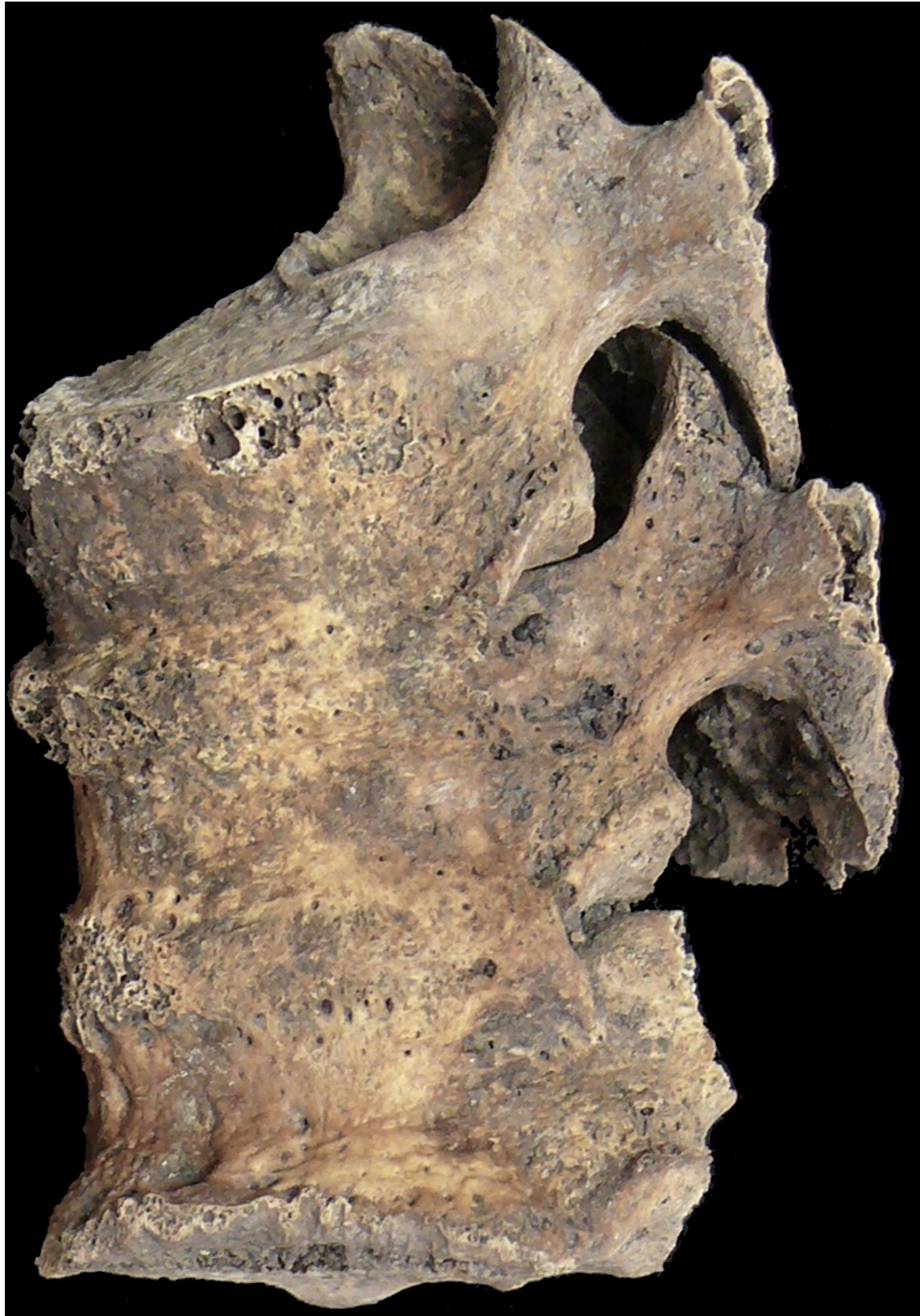

**Fig 3. Fusion of three thoracic vertebrae.**

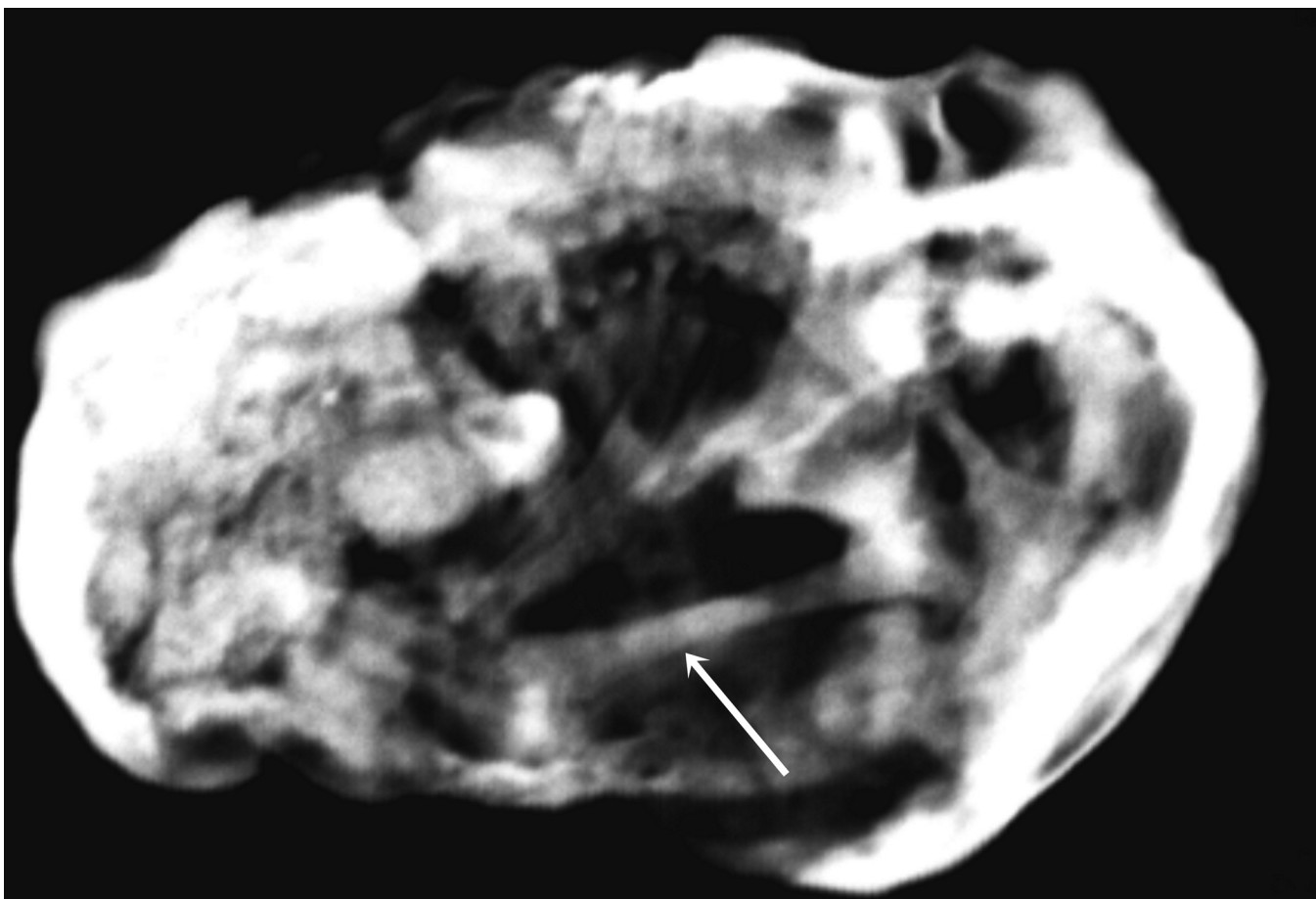

**Fig 4. X-ray imaging of the inside of the cyst.** The arrow marks the femur.

RTG observation (Fig 7). The lengths of femur and tibia shaft obtained, permitted for gestational age estimation. The bone shaft dimensions corresponded with the initial weeks of a second trimester of pregnancy-femur with 16th [46], and the tibia with the 15th [48] or even with the 17–18th week [49].

The histological images show the bone tissue in the early phase of structure organization. Elements characteristic of mature bone, such as osteons, are imperceptible, and the trabeculae are delicate. Trabecular tissue is poorly organized in the body region. The ossification core was noted to be present, as well as areas of high collagen density (Fig 8). The ratio of the bone trabecular area to the intertrabecular area in the region below the ossification line in the epiphyses (BV/TV) was established at under 20%.

The EDS analysis illustrated the presence of elements such as: aluminum, calcium, carbon, copper, fluorine, iron, magnesium, oxygen, phosphorus, potassium, selenium, silicon, sodium and strontium in both bone samples and stone shell (Fig 9). There are noticeable differences, between the samples, in the percentages of the identified elements (Table 1).

## Discussion

Differential diagnosis of the cyst established two possible gynecological cases: *fetus-in-fetu* or *lithopedion*. Taking its macroscopic features into account (eggshell shape, size, the lack of

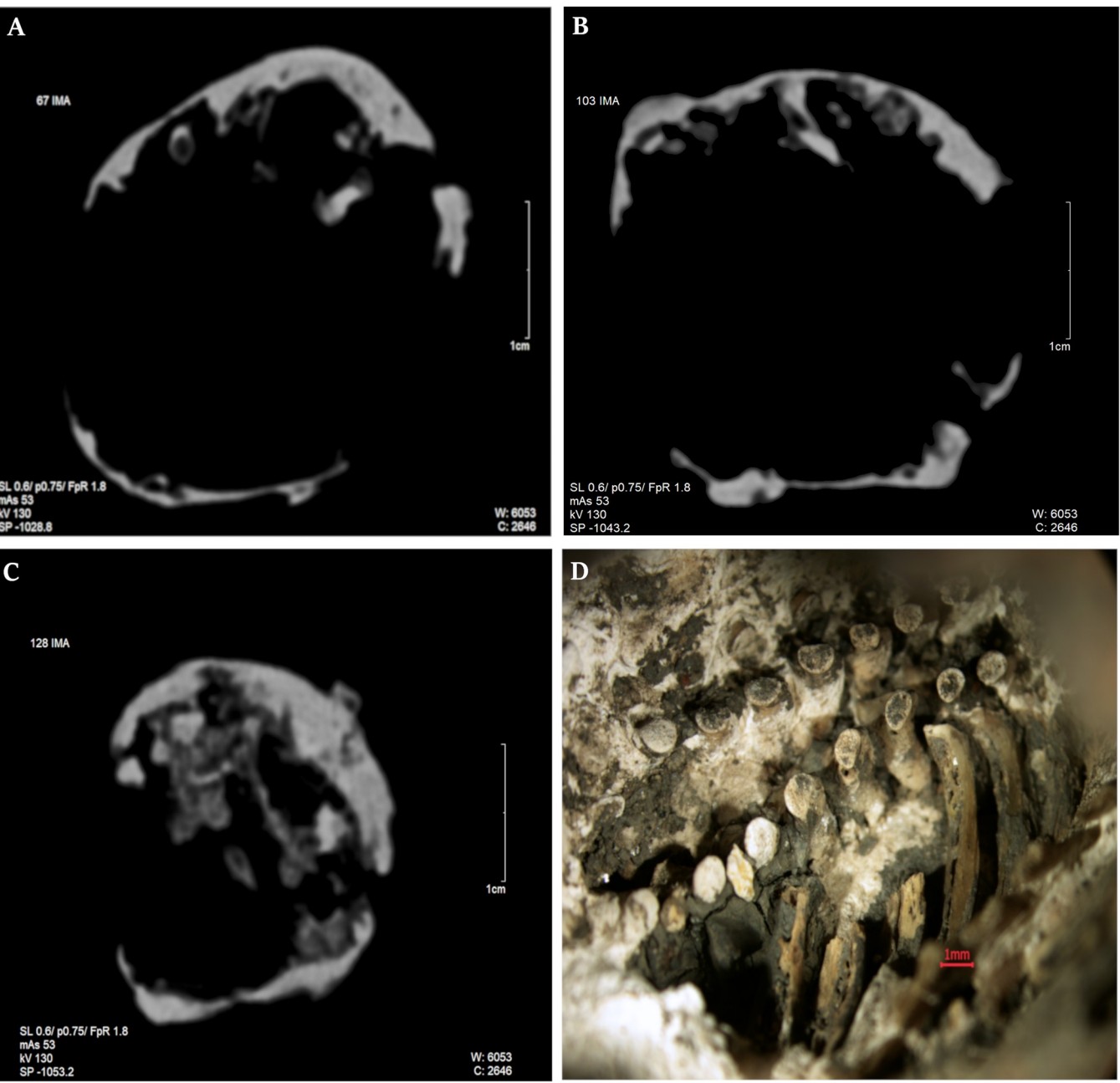

**Fig 5.** Selected CT layers (A-C; slice no. 67, 103, 128 respectively) and inner view of the cyst (D).

vascularization, presence of bone-shaped structures inside) other options for the object etiology, such as urinary or renal calculus, were rejected [hollow inside; 7, 50].

Given the possibilities for being mistaken for stones and discarded or going unnoticed, stone cysts are rare archaeological materials [35]. Such masses are usually examples of dystrophic calcification–deposit of calcium salts (calcium phosphates and calcium carbonates) in degenerated or dead tissues [11].

As a result of their morphological and radiological appearance, they are clinically classified across several types [51, 52]. In the medical and archaeological literature some diagnostic traits

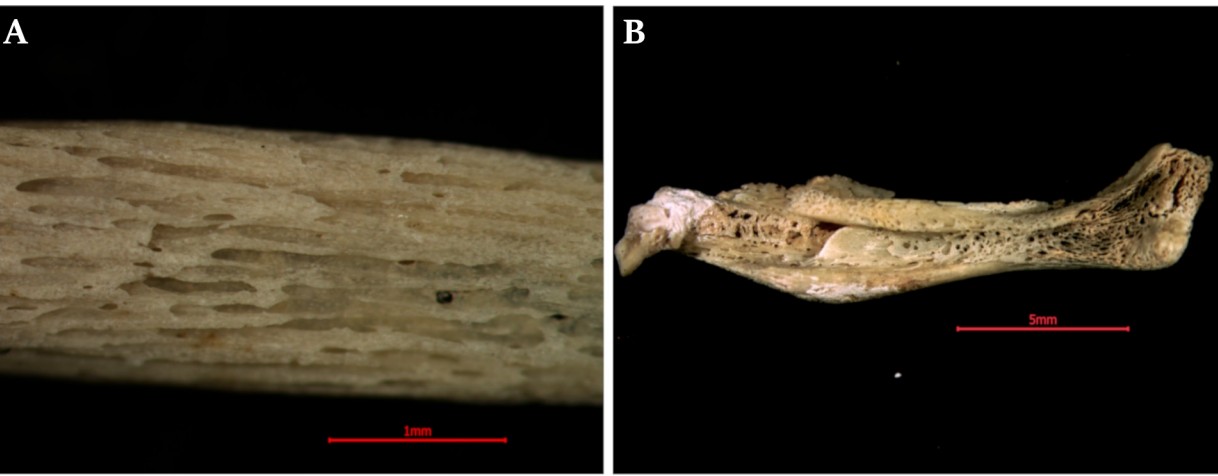

**Fig 6.** Microscopic view of the rib surface (A) and scapula (B).

are useful in differential diagnosis [11]. Drawing on the sex and age of the individual (old woman), localization of the cyst (pelvic area) and its morphology (egg-shell object with bone-like structures inside), three possible explanations were considered in relation to the origin of this phenomenon: i) *fetiform teratoma* (FT), ii) *fetus-in-fetu* (FIF) or iii) *lithopedion*.

Diagnosis of FIF requires a number of criteria to be met [53]. Most importantly, by contrast with *fetiform teratoma*, *fetus-in-fetu* forms an anatomically correct spine, which is the basis for distinguishing between these conditions [54]. An anatomically correct spine was observed in the Libkovice case. Moreover, FIF is usually an anencephalic formation with vestigial limbs–the lower limbs are more developed than the upper limbs [55]–which corresponds with the object in question. Hence, since the analyzed cyst possesses a distinct vertebral structure, the hypothesis of *fetiform teratoma* was rejected.

Five circumstances are required for the formation of *lithopedion*: 1) lack of detection, 2) asepsis fetus, 3) presence of conditions allowing for the deposit of calcium salts, 4) survival of

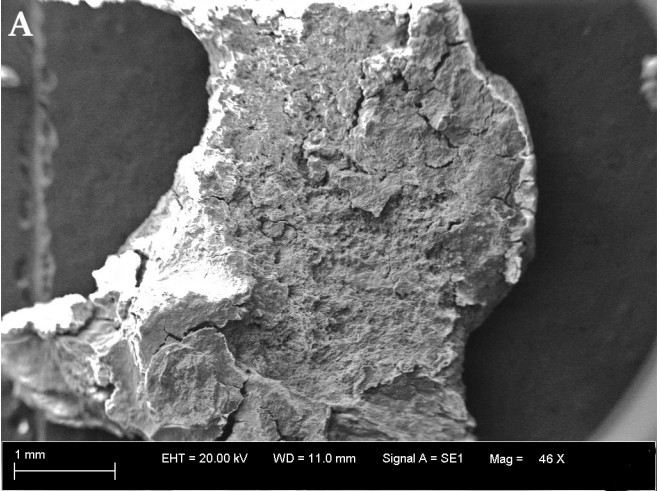

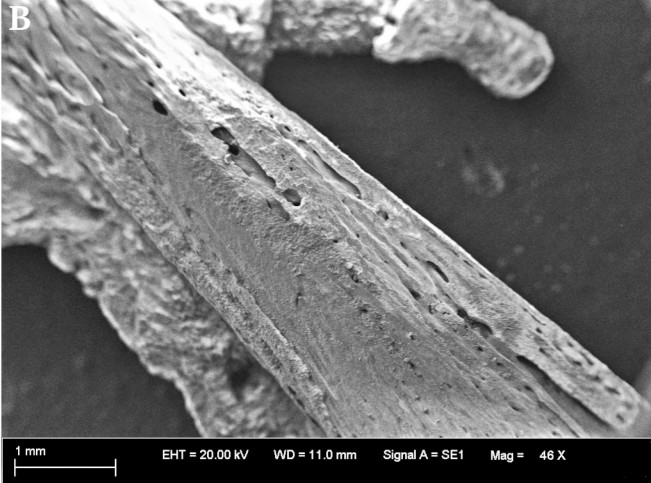

**Fig 7.** SEM images of the shell's inner surface with irregular structures and surface porosity (A) and tibia surface with visible compact tissue and cancellous tissue (B).

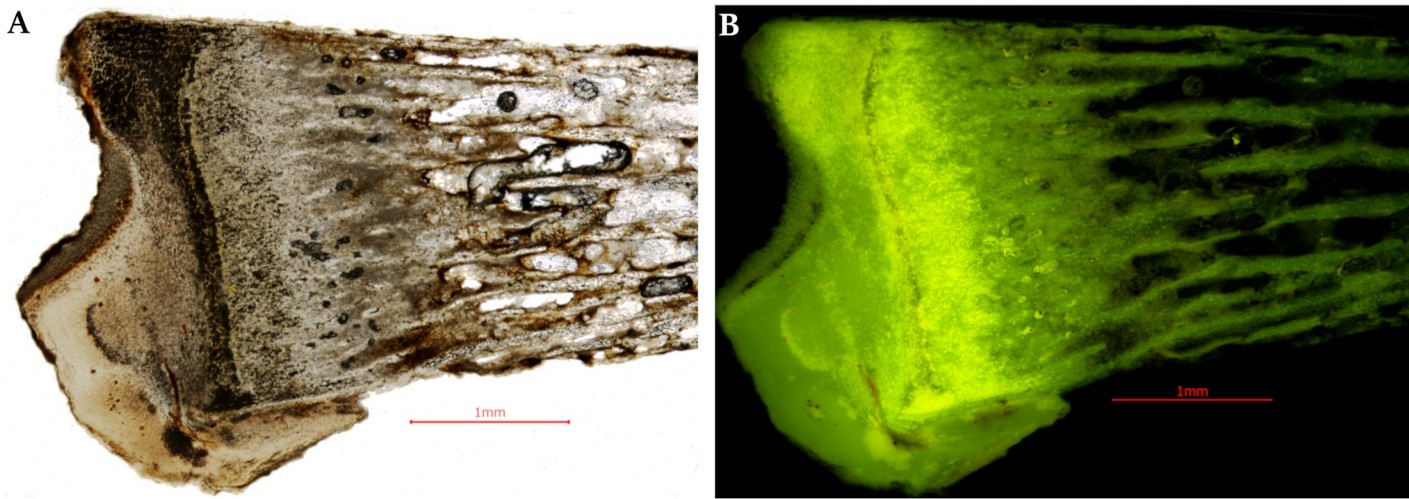

**Fig 8. Sagittal section of the tibia–proximal epiphysis.**

the fetus past the first trimester [28, 56, 57], 5) extrauterine pregnancy [33]. The mean age of women at the time of *lithopedion* detection is 55 years, and the range: 30–100 y. [27], whilst the range within which it is revealed is 4–60 y. [21, 26, 58]. The shortest known period of full calcification of the fetus is 14 months from conception, although it usually takes years until its discovery [33].

The results of histological analyses of bone sample (especially the collagen concentration and BV/TV ratio) confirmed that the discovered object is a human fetus in the second trimester of prenatal development [59–61]. The X-ray measurement of the femur, as well as microscopic photography and measurements, isolated the tibia shaft from the calcified cyst, allowing the gestational age of the fetus to be estimated. The length of the femur shaft stood at 20.2 mm, which points to a gestational age estimate at the 16th week of pregnancy. The tibia shaft was 16

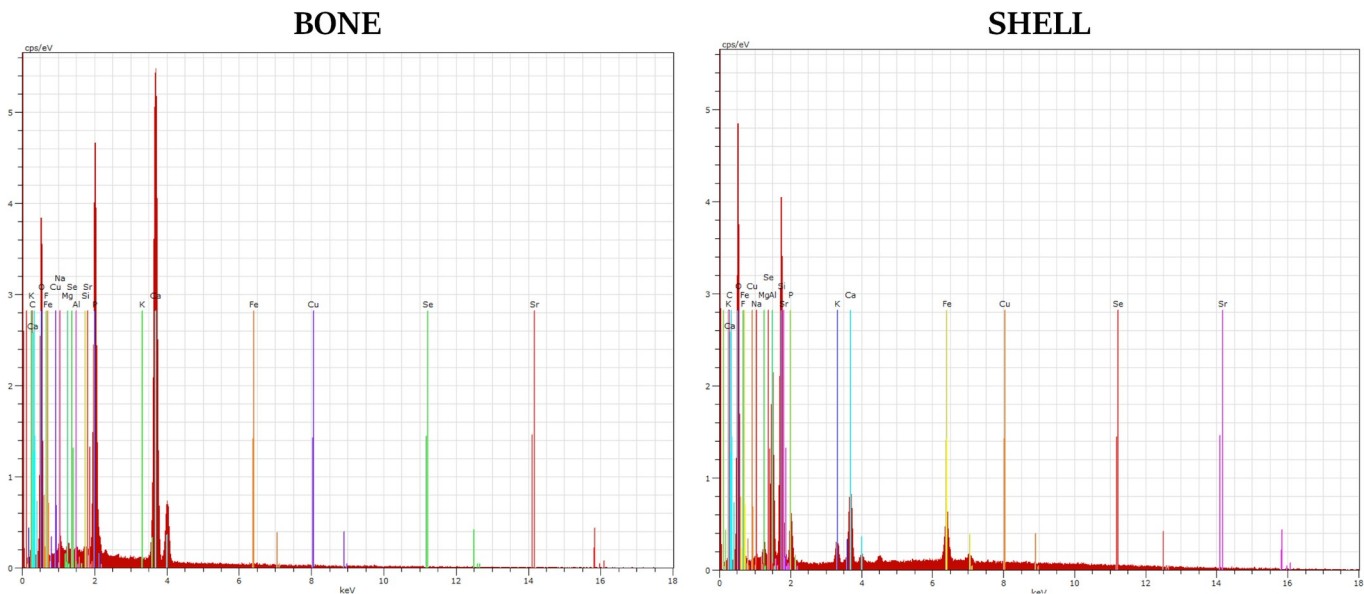

**Fig 9. Percentage composition of the elements in bone and shell.**

mm, which, according to some authors corresponds to the 15[th] week [47], whilst others claim it corresponds with the 17–18[th] [49] weeks of pregnancy. The length of the femur and tibia suggests that the fetus necrosed at the beginning of the second trimester of pregnancy.

The calcium and phosphorus content characteristic of this tissue [62, 63] was found in the bone sample, confirming the assumption it is a biological structure (fetal bone). The level of the oxygen, both in the bone and in the shell, was especially high and likely reflects taphonomic processes [64]. Of particular note, is the very low calcium and phosphorus content, as well as a high content of elements commonly found in soil, such as silicon and aluminum, which were identified in the sample taken from the cyst shell. The low content of calcium, compared with the shell, may indicate that this structure was not hardened in the woman's body during her lifetime. Thus far, this does not concur with observations for *lithopedion* cases [65].

Differential diagnosis revealed that the object is likely to be either *lithopedion* or *fetus-in-fetu*. In this regard, it is impossible to clearly distinguish between the two in the outlined case. Arguments for the *fetus-in-fetu* hypothesis rest on the absence of head and skull structures, the presence of an anatomically correct spine and, stronger development of the lower limbs over the upper limbs [66]. The *lithopedion* hypothesis rests on the age of the fetus–at least 4–5[th] month, one of five crucial criteria for *lithopedion* forming conditions survival of the fetus past the first trimester [28, 56, 57]–its location in the left iliac fossa and covering of hardened shell. The latter, however, does not indicate significant calcium content, contradicting the calcium salt deposit mechanism of *lithopedion* formation. *Fetiform teratoma* was rejected at the outset of the analyses, due to the presence of an axial skeleton, ribs and long bones, the proportions of which were not distorted. The lack of teeth-like structures, often found in FT cases, was an additional support for these conclusions.

## Conclusions

A *maturus-senilis* female skeleton was discovered at the St. Nicholas Church archaeological site (Libkovice, Czechia) in the 18[th] century horizon of a cemetery. The skeleton contained a stone object in the left iliac fossa. Applying a differential diagnosis to the find, the object has been identified as either a *lithopedion* or *fetus-in-fetu* case. The interior of the cysts revealed numerous small bony fragments, typical of a human body. The lengths of the tibia and femur suggest that the fetus was 17–20 weeks old, implying it had necrosed at the beginning of the second trimester of pregnancy. The EDS analysis, performed on the stone shell and bones, revealed a composition of oxygen and silicon or oxygen, calcium and phosphorus, respectively. Although many features of the object indicate its similarity to the diagnostic criteria of lithopedion, the EDS analysis of the shell indicated that it is not composed of calcium, which is known right now as the basic mechanism of lithopedion formation.

The identified object contributes to a very modest global database for this type of find. Such a discovery could help in understanding the etiology of pregnancy disorders and its frequency across populations in a historical context.

## Author Contributions

**Conceptualization:** Barbara Kwiatkowska, Agata Bisiecka, Łukasz Pawelec, Agnieszka Witek, Joanna Witan, Dariusz Nowakowski, Paweł Konczewski, Petr Lissek, Anna Lipowicz.

**Data curation:** Agata Bisiecka, Łukasz Pawelec, Agnieszka Witek, Joanna Witan, Dariusz Nowakowski.

**Formal analysis:** Barbara Kwiatkowska, Agata Bisiecka, Dariusz Nowakowski.

**Investigation:** Barbara Kwiatkowska, Agata Bisiecka, Łukasz Pawelec, Joanna Witan, Anna Lipowicz.

**Methodology:** Barbara Kwiatkowska, Agata Bisiecka, Łukasz Pawelec, Agnieszka Witek, Joanna Witan, Dariusz Nowakowski, Anna Lipowicz.

**Project administration:** Barbara Kwiatkowska, Agata Bisiecka, Paweł Konczewski, Radosław Biel, Petr Lissek, Anna Lipowicz.

**Software:** Dariusz Nowakowski.

**Supervision:** Paweł Konczewski, Petr Lissek, Pavel Vařeka, Anna Lipowicz.

**Visualization:** Agata Bisiecka, Dariusz Nowakowski.

**Writing – original draft:** Agata Bisiecka, Łukasz Pawelec, Agnieszka Witek, Joanna Witan, Dariusz Nowakowski, Radosław Biel, Anna Lipowicz.

**Writing – review & editing:** Barbara Kwiatkowska, Agata Bisiecka, Joanna Witan, Katarzyna Król, Katarzyna Martewicz, Anna Lipowicz.

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
