## [Decision Letter · Decision Letter 0]

29 Apr 2021

PONE-D-21-07709

Differential diagnosis of a cyst found in an 18th century female burial site at St. Nicholas Church cemetery (Libkovice, Czechia)

PLOS ONE

Dear Dr. Bisiecka,

Thank you for submitting your manuscript to PLOS ONE. After careful consideration, we feel that it has merit but does not fully meet PLOS ONE’s publication criteria as it currently stands. Therefore, we invite you to submit a revised version of the manuscript that addresses the points raised during the review process.

We look forward to receiving your revised manuscript.

Kind regards,

David Caramelli, Ph.D

Academic Editor

PLOS ONE

Journal Requirements:

2. In your manuscript, please provide additional information regarding the specimens used in your study. Ensure that you have reported specimen numbers and complete repository information, including museum name and geographic location.

For more information on PLOS ONE's requirements for paleontology and archaeology research, see https://journals.plos.org/plosone/s/submission-guidelines#loc-paleontology-and-archaeology-research.

3. We note you have included a table to which you do not refer in the text of your manuscript. Please ensure that you refer to Table 1 in your text; if accepted, production will need this reference to link the reader to the Table.

Reviewers' comments:

Reviewer's Responses to Questions

**Comments to the Author**

1. Is the manuscript technically sound, and do the data support the conclusions?

Reviewer #1: Yes

Reviewer #2: Yes

2. Has the statistical analysis been performed appropriately and rigorously? 

Reviewer #1: N/A

Reviewer #2: N/A

3. Have the authors made all data underlying the findings in their manuscript fully available?

Reviewer #1: Yes

Reviewer #2: Yes

4. Is the manuscript presented in an intelligible fashion and written in standard English?

Reviewer #1: Yes

Reviewer #2: Yes

5. Review Comments to the Author

Reviewer #1: Dear Authors,

I really enjoyed to read this paper. I think it's very well written, original and with all the data provided.

I recommend some minor revisions:

- it is possible to establish the age range or the mean age and the stature for the female individual?

- I will also add the brand of the machine utilised for X-Rays analyses

- A 3D virtual reconstruction from the Ct-Scan of the cyst would be very interesting to see (for example a picture with the whole cyst and another one cut in norma sagittalis, so one can appreciate the section)

Reviewer #2: The paper reads very well.

I would recommend:

1) adding "calcified" to cyst in the title

2) pointing more on the diagnosis of lithopedion rather than the other one proposed

3) adding one more general, theoretical reference in the introduction for the workflow of palaeopathological diagnosis and its limitations: Rühli FJ, Galassi FM, Haeusler M. Palaeopathology: Current challenges and medical impact. Clin Anat. 2016 Oct;29(7):816-22. doi: 10.1002/ca.22709.

6. PLOS authors have the option to publish the peer review history of their article (what does this mean?). If published, this will include your full peer review and any attached files.

Reviewer #1: No

Reviewer #2: No

---

## [Author Response · Author response to Decision Letter 0]

18 May 2021

According to Reviewer #1 comments:

1.‘It is possible to establish the age range or the mean age and the stature for the female individual?’

Yes, it was possible to establish more precisely the age of the female individual. Based on the morphological details of the skull (cranial suture obliteration) and pelvis (pubic symphysis shape), as well as degenerative changes observed in the bone material, associated with the aging process (presence of articular cartilage degenerative lesions and degeneration of the vertebrae surface), considering that this individual was from a rural population, we estimated age as 45-55 years old. The individual was 158 cm tall, established by Trotter and Gleser method as well as Nainys method. We added this information in the ‘Anthropological analyses of the skeleton’ section (page 8, line 188-192) of the article.

2.‘I will also add the brand of the machine utilised for X-Rays analyses’

Of course you are right – thank you very much for pointing out this oversight. The machines used for X-Ray analyses were: a) for SEM: SEM, EVO LS15, Zeiss with energy dispersive X-ray analysis (SEM/EDX) applying quantax detector (Brüker) with 10 kV of filament tension; b) for X-Ray imaging: Konica Minolta Regius Model 110s with Siemens Polydoros LX 30 generator; c) for CT: Siemens Somatom Emotion 16 (3rd generation) apparatus. All the information was added to the ‘Methods’ part (page 7, line 173-174 & page 8, line 181-184) of the article.

3.‘A 3D virtual reconstruction from the Ct-Scan of the cyst would be very interesting to see (for example a picture with the whole cyst and another one cut in norma sagittalis, so one can appreciate the section)’

Regarding the inclusion of the three-dimensional reconstruction obtained from CT slices with the article: we agree that it would be an interesting form of object imaging, however, after detailed discussion, we decided that in our case it would not enrich the content of the article. 3D reconstructions are generally performed to reveal a closed object, the structure of which we do not want to break. In this case, we did a CT to view the internal structure of an object that we did not know what it was. After confirming that there were structures resembling human (fetal) bones inside, we extracted them from the inside of the shell to perform the necessary EDS and histological analyzes (which would not have been possible without physical interference with the object). Based on the scans (both in 512 dpi and 1080 dpi resolution), we made a 3D reconstruction, but its quality is not satisfactory (due to the delicate structure of the cyst, the edges of the three-dimensional object were blurred). We believe that this reconstruction does not bring significant informational or aesthetic value to the article, so we preferred to include in the article other images showing the way of our thinking from the general (outer view of burial no. 1095) to the detail (histological stainings and SEM). We hope that you will accept our explanation of this issue and lack of the 3D model does not decrease the value of this article.

According to Reviewer #2 comments:

1.‘I would recommend adding "calcified" to cyst in the title’

We added the word ‘calcified’ to the article's title as suggested.

2.‘I would recommend pointing more on the diagnosis of lithopedion rather than the other one proposed’

Regarding the suggestion to emphasize the greater possibility that the discovered object is an example of a lithopedion: we cannot fully agree with this proposal as we believe that the information we have does not allow us to fully draw such a conclusion. Although many features of the object indicate its similarity to the diagnostic criteria of lithopedion, the EDS analysis of the shell indicated that it is not composed of calcium, which is known right now as the basic mechanism of lithopedion formation. We agree that lithopedion is more often described in the literature and has already been identified in the archaeological context, however – referring to the review article suggested by the Reviewer – the frequency of publications on a given phenomenon does not necessarily have to be related to the frequency (and therefore higher probability) of its occurrence. We decided to write this article in such a form that it does not resolve this issue unequivocally, in the hope that it will cause a discussion in the literature on the possibility of fetus-in-fetu occurrence in archaeological remains or on the possibility of lithopedion’s shell formation by a mechanism other than calcium deposition. We trust that such an approach will contribute to the deepening of the scientific potential. We have provided a brief overview of this explanation in the 'Conclusions' section (page 14, line 327-329) and we hope that the current form of the article will be acceptable.

3.‘I would recommend adding one more general, theoretical reference in the introduction for the workflow of palaeopathological diagnosis and its limitations: Rühli FJ, Galassi FM, Haeusler M. Palaeopathology: Current challenges and medical impact. Clin Anat. 2016 Oct;29(7):816-22. doi: 10.1002/ca.22709.’

We read the recommended article and agree that it provides valuable support for the content of our theoretical introduction. We referred to it in the 'Introduction' section (position 6 in the Bibliography).

---

## [Decision Letter · Decision Letter 1]

22 Jun 2021

Differential diagnosis of a calcified cyst found in an 18th century female burial site at St. Nicholas Church cemetery (Libkovice, Czechia)

PONE-D-21-07709R1

Dear Dr. Bisiecka,

We’re pleased to inform you that your manuscript has been judged scientifically suitable for publication and will be formally accepted for publication once it meets all outstanding technical requirements.

Kind regards,

David Caramelli, Ph.D

Academic Editor

PLOS ONE

Additional Editor Comments (optional):

Reviewers' comments:

Reviewer's Responses to Questions

**Comments to the Author**

1. If the authors have adequately addressed your comments raised in a previous round of review and you feel that this manuscript is now acceptable for publication, you may indicate that here to bypass the “Comments to the Author” section, enter your conflict of interest statement in the “Confidential to Editor” section, and submit your "Accept" recommendation.

Reviewer #1: All comments have been addressed

Reviewer #2: All comments have been addressed

2. Is the manuscript technically sound, and do the data support the conclusions?

Reviewer #1: Yes

Reviewer #2: Yes

3. Has the statistical analysis been performed appropriately and rigorously? 

Reviewer #1: N/A

Reviewer #2: N/A

4. Have the authors made all data underlying the findings in their manuscript fully available?

Reviewer #1: Yes

Reviewer #2: Yes

5. Is the manuscript presented in an intelligible fashion and written in standard English?

Reviewer #1: Yes

Reviewer #2: Yes

6. Review Comments to the Author

Reviewer #1: Dear author/s,

your reply is fine to me. You have made all the requested changes and I agree with you regarding the 3D virtual reconstruction.

Reviewer #2: Thanks for making the requested edits and for explaining your views on the correct diagnostic approach.

7. PLOS authors have the option to publish the peer review history of their article (what does this mean?). If published, this will include your full peer review and any attached files.

Reviewer #1: No

Reviewer #2: No

---

## [Editor Report · Acceptance letter]

25 Jun 2021

PONE-D-21-07709R1 

Differential diagnosis of a calcified cyst found in an 18^th^ century female burial site at St. Nicholas Church cemetery (Libkovice, Czechia) 

Dear Dr. Bisiecka:

I'm pleased to inform you that your manuscript has been deemed suitable for publication in PLOS ONE. Congratulations! Your manuscript is now with our production department. 

Kind regards, 

on behalf of

Professor David Caramelli 

Academic Editor

PLOS ONE